# Mechanical Characterization of the Human Abdominal Wall Using Uniaxial Tensile Testing

**DOI:** 10.3390/bioengineering10101213

**Published:** 2023-10-17

**Authors:** Kyleigh Kriener, Raushan Lala, Ryan Anthony Peter Homes, Hayley Finley, Kate Sinclair, Mason Kelley Williams, Mark John Midwinter

**Affiliations:** School of Biomedical Sciences, Faculty of Medicine, The University of Queensland, Brisbane 4072, Australia; r.lala@uq.net.au (R.L.); r.homes@uq.net.au (R.A.P.H.); m.midwinter@uq.edu.au (M.J.M.)

**Keywords:** abdominal wall mechanical properties, tensile properties, uniaxial tensile testing

## Abstract

It is generally accepted that the human abdominal wall comprises skin, subcutaneous tissues, muscles and their aponeuroses, and the parietal peritoneum. Understanding these layers and their mechanical properties provides valuable information to those designing procedural skills trainers, supporting surgical procedures (hernia repair), and engineering-based work (in silico simulation). However, there is little literature available on the mechanical properties of the abdominal wall in layers or as a composite in the context of designing a procedural skills trainer. This work characterizes the tensile properties of the human abdominal wall by layer and as a partial composite. Tissues were collected from fresh-never-frozen and fresh-frozen cadavers and tested in uniaxial tension at a rate of 5 mm/min until failure. Stress–strain curves were created for each sample, and the values for elastic moduli, ultimate tensile strength, and strain at failure were obtained. The experimental outcomes from this study demonstrated variations in tensile properties within and between tissues. The data also suggest that the tensile properties of composite abdominal walls are not additive. Ultimately, this body of work contributes to a deeper comprehension of these mechanical properties and will serve to enhance patient care, refine surgical interventions, and assist with more sophisticated engineering solutions.

## 1. Introduction

Entry into the abdomen is a critical step in abdominal laparoscopic procedures. Although routinely performed in operating theaters across the world, entry into the abdomen is not without risk. Events such as injury to aorta, inferior vena cava, iliac vessels, and hollow and solid viscera are a significant complication, and are associated with a high mortality rate [1]. Potential contributing factors to large vessel or bowel injury include inexperienced/unskilled surgeons, inappropriate insertion of the device used to enter the abdomen, and forceful thrust of the entry device into the abdomen [1].

The implementation of procedural skills trainers (referred to as trainers) during surgical training has been proposed to improve safety during laparoscopic abdominal entry [2]. Trainers have been proposed as a method for exposing surgeons to procedures before training surgeons operatively encounter patients. While there is variability in current training curricula, generally, the first time a training surgeon performs a procedure is on a patient. The introduction of trainers may create more skilled surgeons and mitigate the need for procedural practice on patients during early phases of skill acquisition.

Despite the likely benefits of integrating trainers during surgical training, there is a major concern regarding haptic fidelity. Educational trainers with low haptic fidelity (i.e., ones that do not adequately replicate the tactile feedback of a patient’s tissues) may encourage an inappropriate application of force during learned surgical procedures. It has been suggested, therefore, that trainers should be designed based on the mechanical properties of tissues. To the authors’ knowledge, there is no current laparoscopic port placement trainer on the market.

It is presumed that by choosing synthetic materials with the same or similar properties to human tissues, a trainer with high haptic fidelity can be designed [3,4,5]. Whilst there is a myriad of mechanical properties to choose from, it is theorized that tensile properties may be the best starting point [4]. There is a body of literature that contains some tensile properties of the abdominal wall; however, most of the studies are concerned with hernia repair. Studies informing hernia repair [3,6,7,8,9,10,11] are not sufficient in the context of designing a trainer for port entry because they contain (1) compromised tissue, (2) data from select tissues (i.e., not all layers), or (3) incomplete mechanical data (e.g., missing elastic modulus).

The field of biomechanics is a vast field, and several authors have commented on the need for increased mechanical testing and data from human tissues [3,5,12]. In an effort to assess the availability of mechanical properties from human tissues used in the design of trainers, our research group conducted a scoping review [13]. This extensive study surveyed a total of 4378 articles, and a discernible gap in the existing literature regarding the utilization of biomechanical properties of human tissues in the design of trainers was revealed [13]. Furthermore, our scoping review underscored the absence of biomechanical properties for all layers of the abdomen [13].

In this work, we characterize the tensile properties of a partial composite and the tissues that comprise the human abdominal wall. This work presents the tensile properties of the anterior abdominal wall, specifically, the elastic modulus, ultimate tensile strength, and strain at failure. This work contributes to our broad research goal of using human data to inform the design of a surgical skills trainer focusing on entry into the abdomen during laparoscopic procedures.

### Port Placement & Abdominal Wall Anatomy

Before designing a trainer, it is imperative to understand how injury can occur during entry into the abdomen. An appreciation of (1) the entry procedure and (2) the anatomy of the abdominal wall is required to develop an effective trainer.

During laparoscopic entry into the abdomen, insufflation through a primary port is undertaken to separate the viscera from the ventral abdominal wall. This primary port is often placed at the umbilicus using an open or closed technique.

The open (Hasson) technique involves an incision in the umbilicus and dissection of the subcutaneous tissue and fasciae under direct visualization. Stay sutures are placed in the fascial layers for subsequent use in stabilizing the port. The peritoneum is opened (bluntly or by sharp dissection) and a cannula is advanced into the abdomen for insufflation.

The closed technique involves the use of a Veress needle. The linea alba is commonly used for closed entry as it is the only significant structure between the superficial tissues and the parietal peritoneum. The skin and subcutaneous tissue are dissected, and then the skin and fascia are elevated. The Veress needle is passed through the incision to enter the peritoneum with an angled approach. There is tactile feedback as the needle traverses the linea alba, then the peritoneum. After insufflation, the Veress needle is removed, and the primary trocar is placed in the same tract.

Due to the proximity of important structures (e.g., bowel, blood vessels), it is imperative that clinicians appropriately handle the tools described above and shown below (Figure 1). During abdominal entry, clinicians are expected to apply tactile and visual feedback to their knowledge of the anatomy of the abdominal wall to safely penetrate it. A brief description of the relevant clinical anatomy (adapted from Last’s Anatomy (9th Edition)) [14] is provided here; however, the reader is directed to specialist texts for further detail.

The anterior abdominal wall comprises skin, superficial fascia (subcutaneous fat), muscle and fascial layers, and the peritoneum. The tissue comprising the wall and the number of layers changes depending on their location within the abdominal wall. Deep to the skin, the superficial fascia is a dual layer comprising Camper’s fascia (superficial layer of superficial fascia) and Scarpa’s fascia (membranous layer of superficial fascia). These layers of superficial fascia extend beyond the abdomen to the thorax and pelvis.

There are three muscular layers in the abdominal wall, which remain separate laterally and fuse in the midline as the rectus abdominus. The external oblique muscle is the most superficial of the lateral muscles. The muscle arises from the lower eight ribs and becomes aponeurotic by its broad insertion from the iliac crest and anterior superior iliac spine to the pubic tubercle. Aponeurotic fibers with a free upper border running horizontally from the fifth rib to xiphisternum run anterior to the rectus abdominus muscle and attach to other aponeurotic layers to form the linea alba. This aponeurosis therefore contributes to the anterior rectus sheath (described below). The muscle fibers of the external oblique run obliquely caudomedially.

The next muscular layer, deep to the external oblique muscle, is the internal oblique. It originates from the lumbar fascia, iliac crest, and lateral two thirds of the inguinal ligament. Muscle fibers run towards the costal margin to which they attach. The muscle gives way to aponeurotic fibers at the tip of the ninth costal cartilage, and the aponeurosis contributes to the rectus sheath and fuses with other aponeuroses to form the linea alba.

The transversus abdominus is the deepest of the three lateral muscles. It has a broad origin from the costal margin, lumbar fascia, iliac crest, and inguinal ligament. The muscle fibers become aponeurotic and also contribute to the rectus sheath.

The rectus abdominus is a paired muscle running from the pubic symphysis and pubic crest to the fifth to seventh costal cartilages. The two muscles are separated by the linea alba. The rectus abdominus is divided into smaller bellies by tendinous intersections (usually three) that penetrate the superficial layer of the muscle. It is encased by the rectus sheath (Figure 2) which is formed by the aponeuroses of the lateral abdominal wall muscles. At approximately 2.5 cm below the umbilicus, the anterior rectus sheath is structured differently (the arcuate line). Above the arcuate line, the anterior rectus sheath is formed by the fusion of the aponeurosis of the external oblique and the superficial layer of the split aponeurosis of the internal oblique. The posterior layer of the rectus sheath above the arcuate line is formed by the posterior layer of the aponeurosis of the internal oblique and that of the transversus abdominus.

The splitting of the internal oblique aponeurosis forms the semilunar line, which runs from the pubic tubercle to the ninth costal cartilage. In the region superior to the costal margin, the anterior rectus sheath comprises only the aponeurosis of the external oblique. Below the arcuate line, all three aponeuroses pass anterior to the rectus muscle; the posterior rectus sheath is absent. The arcuate line is formed by the transition from the internal oblique aponeurosis laminating to encase the rectus abdominus superiorly to passing with the aponeurosis of the transversus abdominus anteriorly to the rectus muscle. The aponeurotic fibers interdigitate in the midline to form the linea alba. Deep to the rectus abdominus and transversus abdominis is the transversalis fascia, extraperitoneal fat, and the parietal peritoneum.

As described above, trocar placement traverses all the layers of the abdomen in series and therefore encounters the biomechanical properties of each layer separately. Designing a trainer with high haptic fidelity may require the synthetic materials to reflect the individual layers’ mechanical properties. Alternatively, a synthetic material with the bulk properties of the abdominal wall layers as a composite may suffice. This work investigates the tensile properties of both the individual layers of the abdominal wall as well as those of a partial composite sample.

## 2. Materials and Methods

The human tissues collected were the anterior rectus sheath, posterior rectus sheath, linea alba, rectus abdominis muscle, external oblique muscle, internal oblique muscle, transversus abdominis muscle, and parietal peritoneum. These tissues were extracted from a total of 15 cadavers. Five of the cadavers were fresh-never-frozen (FNF) and tissue had been harvested within 72–96 h post-mortem. The ten frozen (FF) cadavers had undergone at least 1 freeze–thaw cycle prior to tissue dissection and extraction. Tissues from FF cadavers were collected after the specimens thawed at room temperature for approximately 24–48 h. The mean age of donors was approximately 81 and 82 years for FF and FNF specimens, respectively. The mean number of days from death to dissection of FNF donors was approximately 4, while the mean number of days between removal from freezing was approximately 45 days after an average of about 2 freeze-thaw cycles for FF specimens. The available donor characteristics are reported in Table 1.

Ethical approval was obtained from the University of Queensland (2021/HE002373) and performed in accordance with the following legislation: Transplantation and Anatomy, 1979 (Qld) [17], Transplantation and Anatomy Regulations, 2017 (Qld) [18], Criminal Code, 1899 (Qld) [19], and Work Health and Safety, 2011 (Qld) [20].

### 2.1. Extraction of Abdominal Wall Tissues

Approaches for tissue dissection were developed with reference to Grant’s Dissector [21] and Gray’s Anatomy [22]. Tissues were extracted unilaterally from a site devoid of previous surgery. A skin incision was made along the midclavicular line extending from the inferior costal margin to approximately 10 cm inferior to the umbilicus. Camper’s and Scarpa’s fasciae were identified and divided sharply to expose the superficial surface of external oblique muscle. The plane superficial to the external oblique was entered bluntly and extended medially to the semilunar line and laterally to raise a 10 cm flap of skin and subcutaneous fascia. The external oblique was divided sharply 1 cm lateral to the semilunar line and separated from the underlying internal oblique by blunt dissection. Blunt dissection was continued laterally to isolate approximately 10 cm of the external oblique muscle. The external oblique was divided superiorly and inferiorly so that it could be reflected laterally to allow access to the internal oblique. The above dissections were repeated for the internal oblique and transversus abdominus muscles. Following the isolation of the anterolateral muscles, a 5 × 5 cm tissue sample was extracted from each. Finally, a 5 × 5 cm section of peritoneum (with transversalis fascia) was extracted from the lateral abdominal wall deep to transversus abdominus. In vivo orientation muscle samples were labeled and prepared for subsequent analysis.

Next, the anterior rectus sheath was divided medial to the semilunar line and above the umbilicus. Fibrous attachments to the tendinous intersections were divided sharply, and the anterior rectus sheath was reflected medially towards the linea alba. A section of the rectus abdominus (between tendinous intersections) was lifted off the posterior rectus sheath. The anterior rectus sheath, rectus abdominus, posterior rectus sheath, musculofascial composites, and linea alba were sampled.

All tissues were kept in their in vivo orientation and cut into rectangular samples in the transverse and longitudinal directions (Figure 3). Where possible, multiple samples of the same tissues were collected. In some instances (e.g., peritoneum or posterior rectus sheath), there were adhesions, which made it difficult to extract individual tissue layers. Table 2 summarizes the total number of samples taken from each tissue type.

In addition to individual tissues, a tissue composite (referred to as composite) was collected from three different cadavers. The composite contained 4 different tissue layers from the abdominal wall: the anterior rectus sheath, rectus abdominis muscle, posterior rectus sheath, and the peritoneum. The composite was collected to assess the bulk tensile properties when the abdominal wall is partially intact.

Tissue samples were adhered to the smooth side of 320 grit paper (3M, Maplewood, MN, USA) using cyanoacrylate (Loctite ^®^, Henkel, Düsseldorf, Germany) to prevent slippage within the grips of the testing machine (Figure 4).

After samples were adhered to grit paper, a vernier caliper was used to measure sample gauge length, width, and thickness (±0.02 mm). Tissue measurements are summarized in Table 3. All mechanical tests were performed at room temperature (18–21 °C) in the GAF, and tissues were kept hydrated with a specimen-wetting solution prepared by the UQ GAF staff.

### 2.2. Tensile Testing

Tensile testing was conducted using a ST-1001 Universal Testing Machine (Salt, Incheon, Republic of Korea). The Universal Testing Machine (UTM) had a maximum load cell capacity of 5 kilonewtons (kN) (±0.0333 newton [N] resolution), and data were recorded with a sample rate of 194 Hz. Samples were positioned in screw grips (Side-Action Grips 5kN, Salt Incheon, Republic of Korea) and tightened to “finger-tightness”.

Since the goal of this research was to select synthetic materials that match tissue properties, tissues were not pre-conditioned [4]. The cross-head speed was selected based on similar work [3] and was conducted at a quasi-static rate of 5 mm/min. Data collection began once tissues were pre-stressed to 0.2 N and were tested to failure. The point of failure was visually inspected and noted as being either upper third, middle third, bottom third, or within the grips (Figure 5). Tissue failure was defined as not occurring in the grips and as either of the following conditions: (1) a 90% drop in the applied load or (2) when the applied load fell to zero.

After testing was completed, load (N) and displacement (mm) outputs from the Light-Salt testing software (Salt, Incheon, Republic of Korea) were collected to generate stress–strain curves.

### 2.3. Tensile Data from Stress–Strain Curves

Using load and displacement outputs from the Light-Salt testing software (version 12.6.0), stress and strain were calculated. An assumption of incompressibility was made, meaning that the materials were not considered to be porous, and volume was conserved [23]. Therefore, engineering stress (*σ*) was calculated by dividing force (*F*) over the original cross-sectional area (*A*_0_) (Equation (1)).
(1)$$\sigma =\frac{F}{A_{0}}$$

The original cross-sectional area (*A*_0_) was assumed to be rectangular and calculated by multiplying sample width (*w*) by thickness (*t*) (Equation (2)).
(2)$$A_{0}=w\times t$$

Engineering strain (*ε*) was calculated by dividing the change in specimen length (Δ*l*) by the gauge length (*l*_0_) (Equation (3)).
(3)$$\varepsilon =\frac{\Delta l}{l_{0}}$$

Once calculated, stress–strain curves for each sample were plotted using MATLAB (R2022b, The Mathworks, Natick, MA, USA). Once plotted, elastic modulus, ultimate tensile strength (UTS), and strain at failure were extracted from the stress–strain curves. Considering that stress–strain curves of biological tissues exhibit non-linear or hyperelastic curves [23], the elastic modulus was calculated as the first linear portion of the stress–strain curve (Figure 6). In biological materials, the toe region (Region I) is the initial, non-linear portion of the stress–strain curve where there is a small amount of stiffness due to crimped collagen fibers [24]. Following the toe region is the linear region (Region II) of the curve [23]. In this study, the elastic modulus was calculated from Region II by selecting two points and calculating the slope between the two points (Figure 6). The ultimate tensile strength (UTS) was calculated as the maximum stress in the portion of the curve in Region III (Figure 6). Strain at failure was selected from the strain value at failure of the material (Figure 6).

After plotting stress–strain curves, the graphical quality of the curves was assessed. The rating system (developed by the authors) is described in Table 4. Curves that were too noisy (rating of 4) could not reliably provide tensile properties and were excluded from the dataset. A total of 10 stress–strain curves were excluded because of this. All excluded data were muscle samples: rectus abdominis (7), external oblique (2), and transversus abdominis (1).

As mentioned in Section 2.3, the location of failure was noted for tissue specimens. The elastic modulus for specimens that failed in the grips was calculated. However, because the specimens did not meet failure criteria, UTS and strain at failure could not be calculated.

### 2.4. Data Analysis

Tensile properties from stress–strain curves were collated within a master spreadsheet using Excel^®^ (Version 2210, Microsoft ^®^, Redmond, WA, USA). Shapiro–Wilk tests for normality were conducted (α = 0.05). Since the data were non-parametric, Mann–Whitney U tests were used to compare FF and FNF tensile properties for each tissue type (α = 0.05). Statistical testing was performed in SPSS^®^ (29.0.0.0, IBM^®^, Armonk, NY, USA). Scatter and bar charts were created in MATLAB (R2022b, The Mathworks, Natick, MA, USA).

## 3. Results

A total of 91 fresh-never-frozen (FNF) and 138 fresh-frozen (FF) tissues were extracted from the abdominal walls of 15 different cadavers. Representative curves from each tissue type and a combination sample are displayed in Figure 7. As demonstrated in Figure 7, when tested in uniaxial tension, the tissues of the abdominal wall exhibit non-linear elastic stress–strain curves. Toe regions were observed in 158 samples. The median strain value where the toe region ends was 14.05%. The median strain for the end of the elastic region was 37.52%.

### 3.1. Inter- and Intra-Cadaveric Differences in Tensile Properties

As described in Section 2.3, the elastic modulus, ultimate tensile strength, and strain at failure were obtained from the stress–strain curves for each sample. A swarmchart (Figure 8) shows the spread of the mechanical properties between and among cadavers.

As seen in Figure 8, there is a larger spread of elastic moduli and ultimate tensile strength values for the fascia (anterior/posterior rectus sheaths and linea alba) and peritoneum, whereas the tensile properties of the various muscles seem to be less variable. The intra-cadaveric tissue variability of fascia (anterior rectus sheath) versus muscle (external oblique) is again illustrated in Figure 9. Figure 9 demonstrates the broader range of values for tensile properties from anterior rectus sheath samples (elastic modulus: 0.95–6.88 Megapascals [MPa]; UTS: 0.82–1.79 MPa; strain at failure: 52.79%–234.45%) compared to samples from the external oblique muscle (elastic modulus: 0.21–0.59 MPa; UTS: 0.23–0.42 MPa; strain at failure: 93.28%–251.95%).

### 3.2. Fresh-Never-Frozen (FNF) and Fresh-Frozen (FF) Tensile Properties

Tensile property data from different cadaveric preservations (i.e., FNF and FF) were collated by tissue type (Table 5). The median and interquartile ranges (IQRs) for the data are presented because the data are non-parametric. As seen in Table 5, the median values for the elastic moduli of fresh-frozen tissues were generally stiffer than those of FNF cadavers. The three exceptions were the posterior rectus sheath, peritoneum, and transversus muscle. UTS was generally higher for FF tissue except for the posterior rectus sheath and transversus muscle. Values for strain at failure were higher in FNF cadavers for five of the tissue types (anterior rectus sheath, posterior rectus sheath, peritoneum, internal oblique muscle, and composite).

Tissue tensile properties by cadaveric preparation were assessed by the Mann–Whitney U test. The median elastic modulus was significantly higher in the external oblique of FF samples (0.57 vs. 0.24 MPa, *p* = 0.009), and the median ultimate tensile strength was significantly higher in the external oblique of FF samples (0.28 vs. 0.16 MPa, *p* = 0.015). A significant difference in medians was also detected for UTS of the rectus abdominis (FF significantly greater than FNF [0.14 vs. 0.06 MPa, *p* = 0.014]). However, it should be noted that in the case of rectus abdominis comparisons, there was a small sample size for FNF tissues.

### 3.3. Composite Tissue vs. Separated Tissues

The composite contained tissue layers in the following order: anterior rectus sheath, rectus abdominis muscle, posterior rectus sheath, parietal peritoneum. As previously detailed in Table 2 (Section 2.1), a total of seven composite samples were extracted from three different cadavers. The tensile properties of the composites of each cadaver were compared to the tensile properties of the tissues that comprise the composite (Figure 10). Interestingly, when comparing the composite tissue to that of its individual parts, the tensile properties do not seem to have an easily discernable pattern.

## 4. Discussion

The tensile properties generated in this study will be used to select synthetic materials with the same properties in the design of a laparoscopic trainer. Although the generated data have a specific purpose for our research, there is potential that the data can be used in many other engineering applications. For example, the data can also be used for selecting parameters in dynamic tensile testing (e.g., creep-relaxation testing), computational modeling parameters (e.g., finite element analysis), and the design of medical devices/materials (e.g., sutures that match the mechanical properties of the abdominal wall).

In this study, the mechanical characteristics of anterior abdominal wall tissues were investigated. Uniaxial tensile tests were conducted for eight different tissue types and a partial composite, and tensile properties were extracted from the plotted stress–strain curves. While there was variability in the shapes of the stress–strain curves, generalizations can be made. All of the curves could be classified as non-linear elastic or hyperelastic. Most curves had an initial toe region, in which the tissue sample could withstand an appreciable amount of strain without a large change in stress. Following the toe region, a linear region was observed. Generally, this region was found within a strain range of 14%–42%. The toe and elastic region findings are consistent with what has been reported in the literature for similar tissues [6].

Following the linear region, two generalizations about the stress–strain curves in the failure region could be made: (1) the linear region led to a singular peak in the curve, or (2) the linear region led to a series of peaks (usually two), with the second peak being higher. Similar findings have also been reported in the literature [7]. After the maximum stress (UTS) in the stress–strain curves, the curves generally had a dramatic drop in stress followed by a gradually decreasing slope until the definition of failure was met. As previously reported in similar tissues, throughout the curve, there are multi-picks, which are likely due to strips of tissue rupturing until the specimen meets the definition of failure [6,7].

Like the stress–strain curves, variability in the mechanical properties of human tissues has been reported within and between cadaveric donor specimens [6,7]. This variability is likely due to several factors such as the non-homogenous microstructure of tissues, tissue hydration, and donor-specific factors (e.g., age, pathology, sex, etc.) [7,9,25,27,28]. Unfortunately, we cannot explore the specific contributions of the aforementioned factors without corresponding video analysis from the mechanical testing, chemical analysis, and/or histological testing.

### 4.1. Mechanical Properties of the Anterior Abdominal Wall

To the authors’ knowledge, this is the first article that reports on the mechanical characteristics of different tissue types and composites from cadaveric abdominal walls, utilizing uniaxial tensile testing. In this study, a total of 229 abdominal wall tissue samples from 15 different cadavers were tested in uniaxial tension at a quasi-static rate until failure.

In this study, we assessed the mechanical properties of the four anterior abdominal muscles: external oblique, internal oblique, transversus abdominus, and rectus abdominus. Th median elastic moduli of the individual muscles closely align with values published by Cardoso [3] (values given as our median vs. mean from Cardosa): rectus abdominis (0.26 MPa vs. 0.52 MPa), external oblique (0.41 MPa vs. 1.00 MPa), internal oblique (0.53 MPa vs. 0.65 MPa), and transversus abdominis (0.95 MPa vs. 1.03 MPa). The UTS values for the individual muscles were also comparable to work conducted by Cardosa [3] (values give as our median vs. mean from Cardosa): rectus abdominis (0.13 MPa vs. 0.23 MPa), external oblique (0.25 MPa vs. 0.57 MPa), internal oblique (0.32 MPa vs. 0.39 MPa), transversus abdominis (0.73 MPa vs. 0.73 MPa). Data for strain at failure were not available in the literature. However, Cardosa [3] reported values for stretch at UTS, which were compared to our values for stretch at UTS (our mean stretch vs. Cardosa’s mean stretch): rectus abdominis (1.52 vs. 1.60), external oblique (1.89 vs. 1.98), internal oblique (1.87 vs. 1.94), and transversus abdominis (1.79 vs. 2.19).

The tensile properties for the anterior rectus sheath from this study were also similar to the values available in the literature for elastic modulus and UTS. The elastic modulus from this study (6.98 MPa) fell in the range reported in the literature (5.6 MPa [6]–9.06 MPa [3]), as did our UTS value (2.70 MPa) vs. (2.2 MPa [6]–2.86 MPa [3]). However, our strain at failure (128%) was dissimilar to the values available in the literature (26.09%–6.62%) [10]. It is unclear if this is due to the definition of failure or differences in methodology, as neither are well defined in the study from which the comparable data were extracted [10].

The elastic modulus of the linea alba from this study (3.92 MPa) fell within the values reported in the literature; however, this was noted to be a wide range (8 MPa–72 MPa [11]). The wide range in values from the literature is a result of the directionality (i.e., anisotropy) of testing (longitudinal vs. transverse). When the data from this study were filtered to match the testing properties of the comparable study [11] (i.e., data from fresh-frozen cadavers and moduli from transverse and longitudinal directions instead of collated), our average elastic modulus values followed a similar pattern of a larger elastic modulus in the transverse direction (9.54 MPa) compared to the longitudinal direction (3.79 MPa). However, the values from this study were still lower than those reported in the literature [11]. Values for UTS and strain at failure were not available in the literature.

Previous work on the posterior rectus sheath and peritoneum is limited. One study [7] reported a range for the UTS of the transversalis fascia, which contributes to the posterior rectus sheath. The transversalis fascia obtained in that study was obtained from below the arcuate line and in the inguinal canal [7]. In contrast, this study obtained the posterior rectus sheath from above the arcuate line from behind the rectus abdominis muscle. The posterior rectus sheath obtained in this study would have comprised fibers from the aponeuroses of the external and internal oblique muscles. Despite the slight differences in composition and location, the UTS range from the literature (0.63 MPa–1.99 MPa [7]) was similar to the value of the posterior rectus sheath UTS from this study (3.92 MPa).

Ranges for strain at failure are reported for the posterior rectus sheath in two studies: the first is the same study that reported UTS values (i.e., transversalis fascia) [7] and the second is a study that reported values for strain at failure of the anterior rectus sheath [10]. The reported values are 71–104% and 30.77%–35.04%, respectively [7,10]. The literature ranges are lower than the value from this study (133%); however, this is likely due to both differences in definition and methodology.

While there are no available values for the elastic modulus of the posterior rectus sheath, we can conclude that the values from this study would be similar to those obtained by Kirilova et al. [7] based upon similarities in the toe region modulus. Kirilova et al. calculated a secant modulus in the toe region (strain of 5–10%) and found the toe modulus to be approximately 2.82 MPa–8.42 MPa [7]. Utilizing the same methodology, the toe modulus from this study was found to be approximately 13.37 MPa.

In this study, the composite studied comprised the anterior rectus sheath, rectus abdominis muscle, posterior rectus sheath, and peritoneum. It was difficult to readily identify a pattern in the mechanical properties of the composite vs. the individual tissues comprising the composite. Although human studies of composites were not identified, two animal studies assessing bilayer abdominal wall tissue mechanics were found [29,30]. In general, the literature findings were similar to ours: (1) separated layers had different mechanical properties to the composite, (2) large strains were observed, and (3) the composite was less compliant than its corresponding muscle layer. These findings are likely due to the difference in the arrangement of the extracellular connective tissue matrix and muscles; therefore, composites are likely to behave in an intermediate way than their separated tissues [30]. The phenomenon of composites having intermediate properties was described as “myofascial force transmission” [30] and is something to be considered in the future testing of composites.

### 4.2. Study Strengths and Limitations

This study presents novel findings, specifically, tissue stress–strain curves, mechanical properties of abdominal wall tissues, and mechanical testing of a partial abdominal wall composite. In this study, we aim for repeatability by presenting a thorough description of our methods. The current testing conditions were adapted from those previously published [15] and generated comparable results, suggesting these protocols can be replicated.

This study also reports mechanical values for fresh-never-frozen (FNF) and fresh-frozen (FF) cadavers. Comparison of FNF and FF cadavers (especially soft tissues) remains to be investigated; however, it is generally assumed that their mechanical properties are similar. This work provides preliminary evidence of slight differences in tensile properties; however, further analysis is required. Future analyses studying the effects of cadaveric preservation on soft tissue tensile properties are planned.

Some weaknesses of this study include cadaver characteristics, general methodology, and absent reporting on anisotropy. The use of cadavers to study human tissues is essential for destructive mechanical testing such as tensile testing. However, as is the case in this study, donors are generally of advanced age. The effects of aging and pathology on human tissues are likely to be greater and skew the data towards increased stiffness. Additionally, mechanical properties may differ between in vivo tissues and those tested ex vivo. The resultant discrepancy between the properties tested and those of in vivo tissue may impact the design of trainers and should be considered in future studies.

In this study, the methodology has been reported in detail, but some adjustments will be applied in future work. We did not take tissue measurements after pre-stressing the tissues. Tissue is known to be pre-stressed, and future protocols will consider a similar method used by Hernandez et al. to account for initial strains [29]. Furthermore, values for pre-stress can be extracted from studies using MRI data to calculate physiological stresses [8]. Additional changes to the tensile testing protocol include the use of high-speed cameras for recording slippage at the grips and an extensometer for changes in tissue measurements. The inclusion of both will improve the repeatability of our experiments and reliability of our data. Despite the methodological limitations, the similarity in tensile property values to those in the literature is encouraging.

Finally, we did not characterize the anisotropy of the tissues of the abdominal wall. Human tissues are known to be anisotropic. However, it should be noted that these data have been collected, and analyses are planned. Anisotropy was out of the scope of our initial characterizations of the tissues of the abdominal wall.

## 5. Conclusions

This study presents initial characterizations of tissues from the anterior abdominal wall from 15 different fresh-never-frozen and fresh-frozen cadavers. In this work, representative stress–strain curves and tensile properties for eight different tissues and a partial composite are presented. Most of the work presented here is novel; however, in the few instances where similar values are presented in the literature, we demonstrate that the tensile properties of human tissues are reproducible. Future work will include analyses on the effects of the anisotropy of the tissues of the anterior abdominal wall.

Regarding our overall goal of designing trainers with high haptic fidelity, this study presents initial mechanical values that can be used to select synthetic materials. As observed, human data have a range of values for mechanical properties; therefore, our future studies will focus on the sensitivity of physicians to this range of data. Future studies will also try to assess whether tensile properties contribute to the haptics physicians feel during procedures. Although there is much work to be done in this field of research, the findings from this study reveal exciting opportunities to revolutionize how we approach tactile feedback in the design of procedural training devices.

## Figures and Tables

**Figure 1 bioengineering-10-01213-f001:**
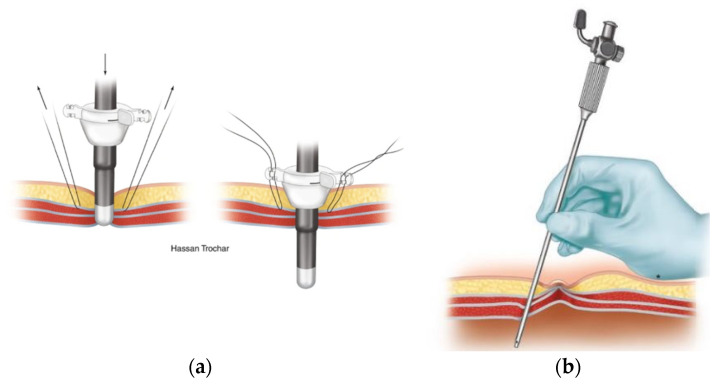
Open and closed port entry are shown in this figure. Panel (**a**) displays an example of open (Hasson) technique for port entry. Image from Figure 1.3, Ref. [15]. Panel (**b**) demonstrates entry using the closed (Veress needle) technique and image from Figure 1.4, Ref. [15].

**Figure 2 bioengineering-10-01213-f002:**
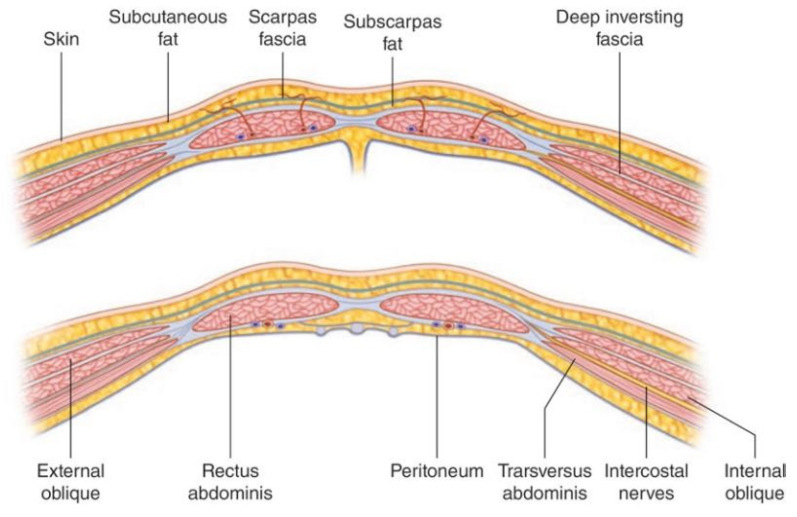
Composition of the abdominal wall is shown (image from Figure 5.4, pg. 45 Ref. [16]). The above image shows the composition of the abdominal wall above the arcuate line, and the image below shows the abdominal wall below the arcuate line.

**Figure 3 bioengineering-10-01213-f003:**
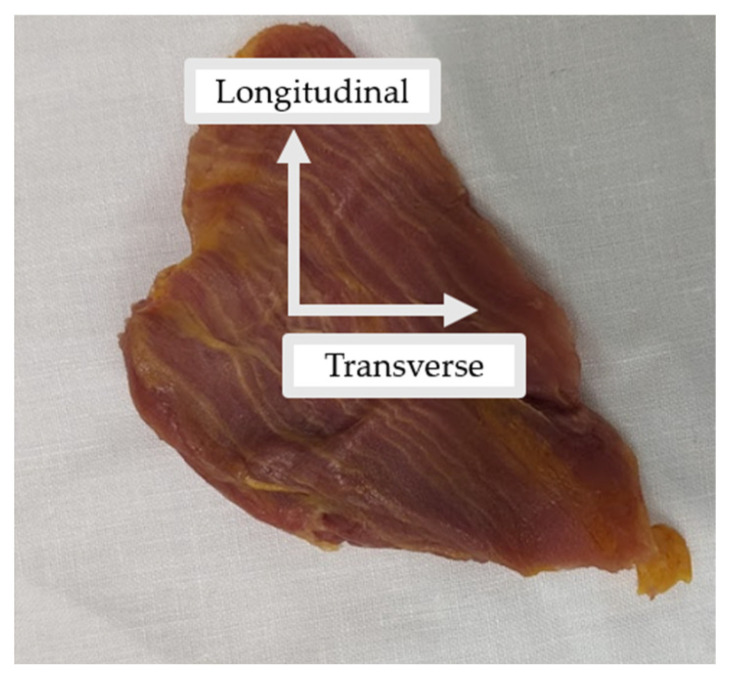
Example of the external oblique muscle extracted in its in vivo orientation. The longitudinal and transverse axes demonstrate the orientation in which samples were cut for mechanical testing. The approximated height and length of the extracted tissue is 5 cm in each direction.

**Figure 4 bioengineering-10-01213-f004:**
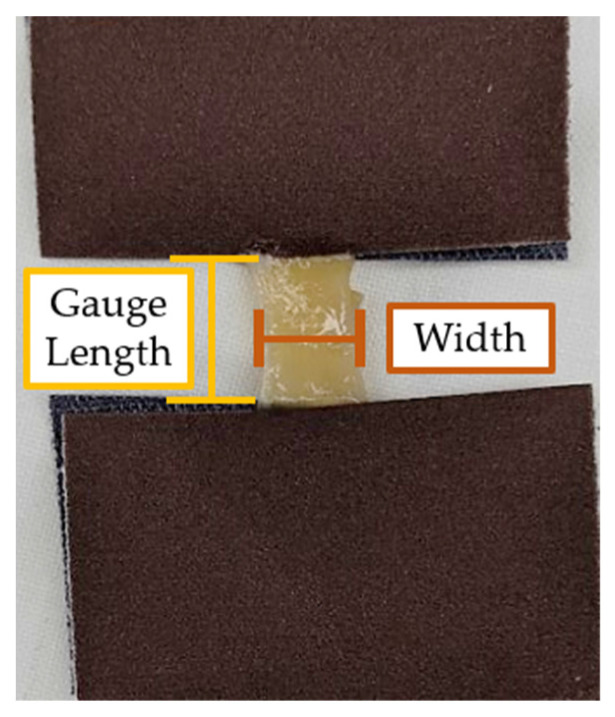
A sample of parietal peritoneum adhered to Grit Paper. An example of locations for gauge length and width measurements are shown in this figure.

**Figure 5 bioengineering-10-01213-f005:**
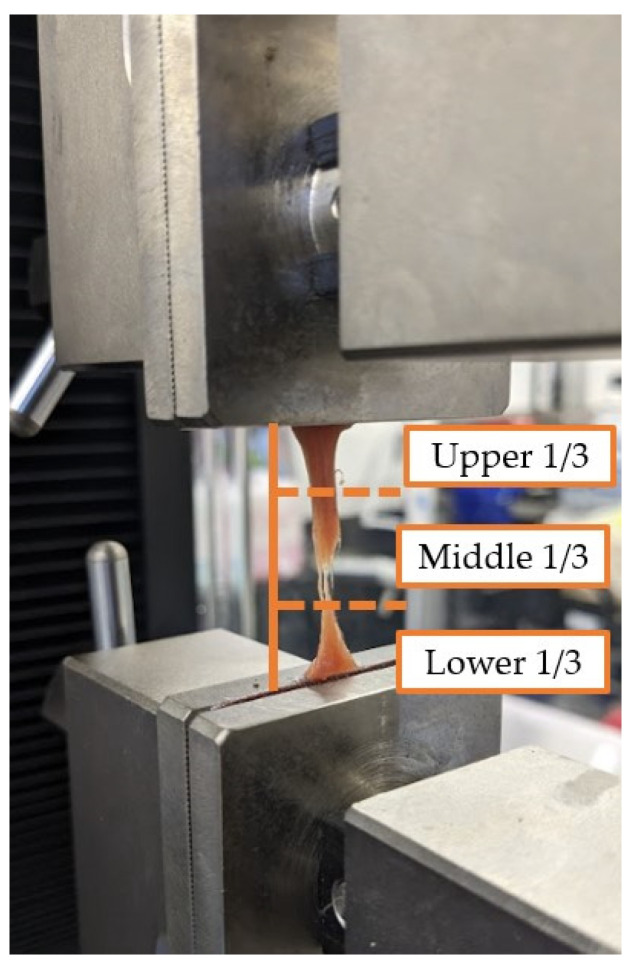
Example of muscle tissue failure within the UTM. The figure also shows the visual approximations of tissue failure. In Figure 5, the sample was noted to have failed in the middle third of the sample.

**Figure 6 bioengineering-10-01213-f006:**
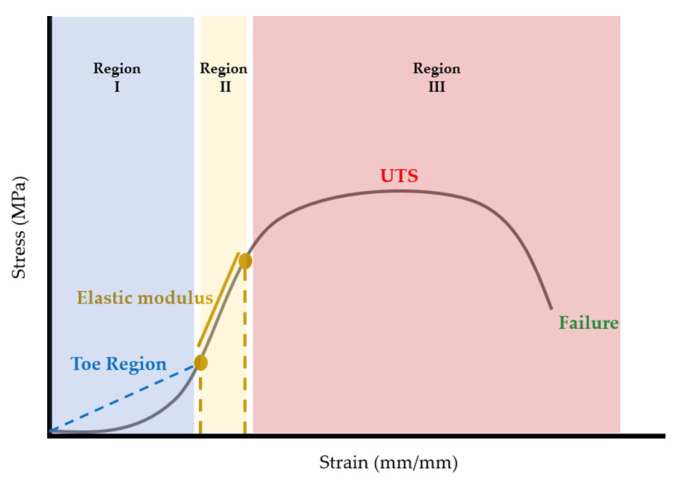
Schematic demonstrating how various tensile properties from the non-linear stress–strain curves were plotted. The stress–strain curve can be broken into three regions [24,25]. The elastic modulus (Region II, gold) was calculated from the first linear portion of the curve following the toe region (Region I, blue). The ultimate tensile strength (UTS) is shown in red, and the failure point is shown in green. Both the UTS and the strain at failure are located in Region III (red).

**Figure 7 bioengineering-10-01213-f007:**
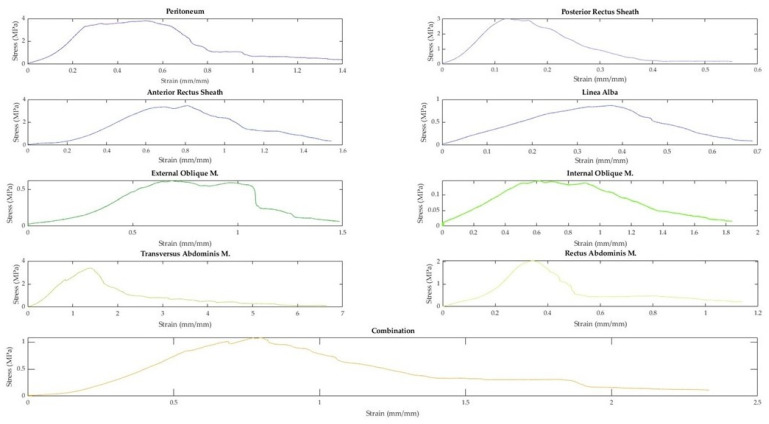
Representative stress–strain curves of abdominal wall tissues.

**Figure 8 bioengineering-10-01213-f008:**
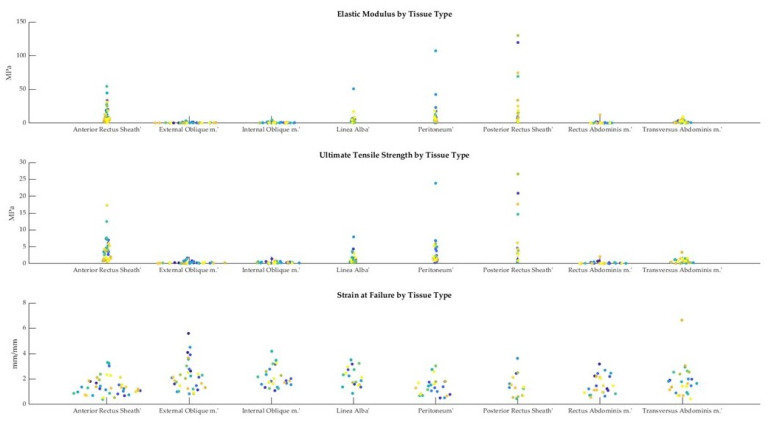
Swarmcharts for tensile properties (elastic modulus, ultimate tensile strength, and strain at failure) are shown for the various tissue types of the anterolateral abdominal wall. Each cadaveric specimen is represented by the same color. Points of the same color indicate multiple tissue samples from the same cadaver. The points of the swarmchart are jittered on the x-axis for ease of visualization [26].

**Figure 9 bioengineering-10-01213-f009:**
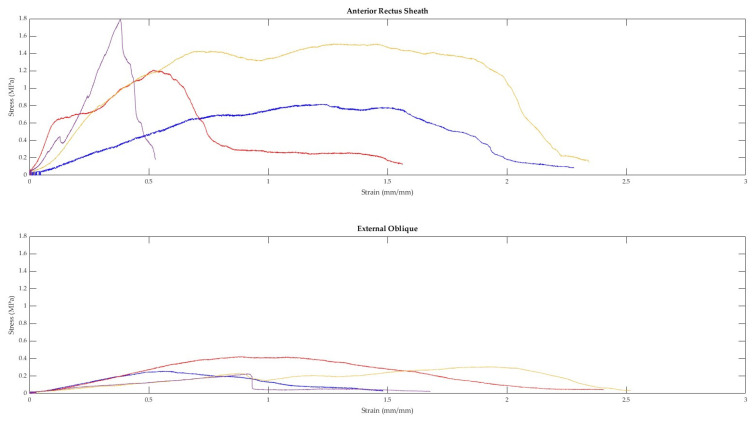
Four anterior rectus sheath and four external oblique muscle samples were taken from the same cadaver and tested in tension. The above figure displays the stress–strain curves from each sample.

**Figure 10 bioengineering-10-01213-f010:**
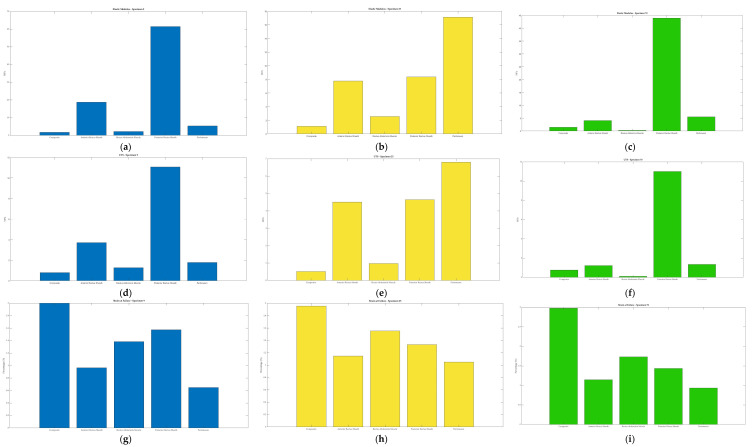
The bar graphs show comparisons of composite samples and their individual tissues (anterior rectus sheath, rectus abdominis, posterior recuts sheath, parietal peritoneum) for elastic modulus (**a**–**c**), ultimate tensile strength (UTS) (**d**–**f**), and strain at failure (**g**–**i**). Samples were taken from three different cadavers: Specimen 2 (blue), Specimen 5 (yellow), and Specimen 13 (green). The x-axes for all bar charts are in the following order: composite, anterior rectus sheath, rectus abdominis muscle, posterior rectus sheath, and peritoneum.

**Table 1 bioengineering-10-01213-t001:** Mean values (standard deviation) of the descriptive characteristics of donors used in this study are presented in the table below. Data are separated by fresh-frozen (FF) and fresh-never-frozen (FNF) specimens.

	Sample Size	Age (Years)	Number of Days between Death and Dissection	Number of Freeze–Thaw Cycles	Number of Days between Last Thaw and Dissection
FF	10	81.1 (7.43)	155.5 (103.5)	2.2 (0.40)	45.3 (98.4)
FNF	5	82.3 (13.3)	4.25 (0.829)	—	—

**Table 2 bioengineering-10-01213-t002:** Summary of the tissues collected from each specimen (1–15).

	1	2	3	4	5	6	7	8	9	10	11	12	13	14	15	Total
Anterior Rectus Sheath	2	2	2	2	2	2	4	2	4	2	2	4	4	4	4	42
Posterior Rectus Sheath	—	2	—	—	2	2	—	—	1	2	2	3	2	2	2	20
Linea Alba	—	2	2	2	—	2	2	2	2	2	2	—	—	—	4	22
Peritoneum	—	2	2	2	2	1	3	2	1	2	1	1	3	—	4	26
Rectus Abdominis	4	2	2	2	2	2	2	—	2	1	1	1	4	3	4	32
External Oblique	3	2	2	2	2	2	—	2	4	2	2	3	4	—	4	34
Internal Oblique	—	2	2	2	2	2	2	2	2	2	2	—	4	—	4	28
Transversus Abdominis	—	2	1	—	2	2	2	2	2	2	2	3	4	—	4	28
Composite	—	2	—	—	2	—	—	—	—	—	—	—	3	—	—	7
Total	9	18	13	12	16	15	15	12	18	15	14	15	28	9	30	239

**Table 3 bioengineering-10-01213-t003:** Medians (interquartile range [IQR]) of length, width, and thickness are summarized in the table below.

Tissue	Sample Size	Gauge Length (mm)	Width (mm)	Thickness (mm)
Anterior Rectus Sheath	42	14.25 (12.43)	7.22 (3.73)	0.59 (0.38)
Posterior Rectus Sheath	20	12.17 (7.98)	5.95 (2.38)	0.32 (0.39)
Linea Alba	22	7.87 (4.57)	5.90 (2.52)	0.88 (0.76)
Peritoneum	26	12.29 (7.31)	5.82 (2.35)	0.39 (0.29)
Rectus Abdominis	25	13.14 (4.66)	6.66 (2.48)	1.82 (0.74)
External Oblique	33	14.02 (7.31)	7.78 (3.96)	2.26 (1.28)
Internal Oblique	28	12.78 (6.74)	6.64 (2.22)	2.07 (0.92)
Transversus Abdominis	26	12.47 (6.57)	6.52 (3.47)	0.93 (0.53)
Composite	7	8.90 (2.50)	7.36 (1.36)	2.40 (0.48)

**Table 4 bioengineering-10-01213-t004:** Stress–strain curve rating system, descriptions, and curve examples.

Rating	Description	Example
1	Very little noise Tensile properties (e.g., toe region, linear elastic region, and ultimate tensile strength) are easily discernible	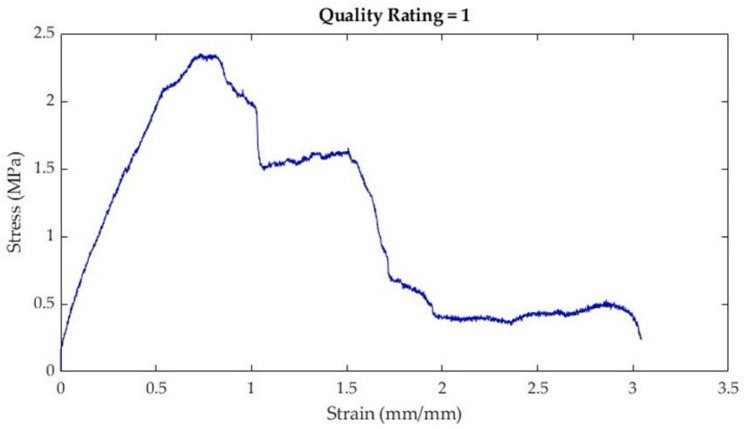
2	Some noise Tensile properties (e.g., toe region or linear elastic region) are visible, but not as easy to discern	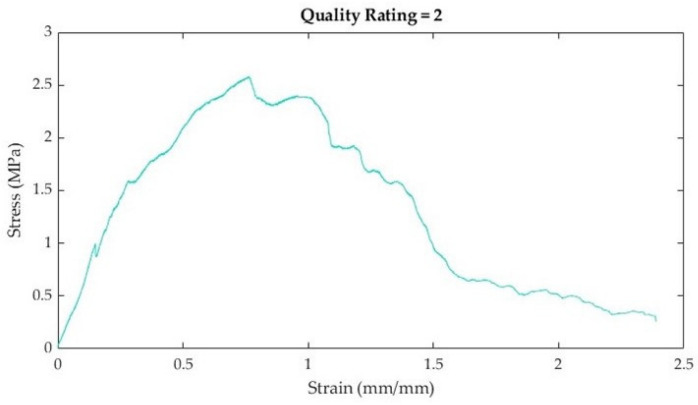
3	Quite a bit of noise Linear elastic region can still be estimated	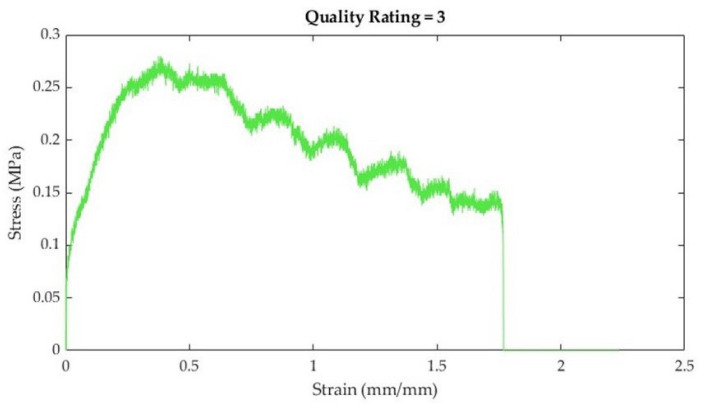
4	Lots of noise Too difficult to discern tensile properties with a high degree of reliability	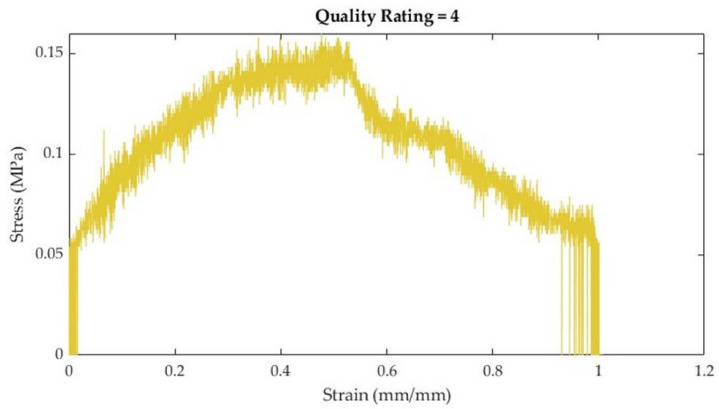

**Table 5 bioengineering-10-01213-t005:** Median and (interquartile ranges [IQRs]) are displayed below for separate tissues and as a composite. Values are given for fresh-never-frozen (FNF) and fresh-frozen (FF) cadavers. Outputs from the Mann–Whitney U test are included in the footer of this table.

Cadaveric Preparation	Tissue	Elastic Modulus (MPa)	Ultimate Tensile Strength (MPa)	Strain at Failure (%)
FNF	Anterior Rectus Sheath	6.02 (10.14)	1.54 (3.02)	132.92 (139.00)
	Posterior Rectus Sheath	14.32 (57.62)	4.14 (13.34)	134.92 (116.00)
	Linea Alba	2.67 (6.87)	0.87 (1.28)	195.96 (126.00)
	Peritoneum	6.79 (6.09)	1.80 (3.99)	145.54 (118.00)
	Rectus Abdominis	0.19 *	0.06 *^§^	92.21 *
	External Oblique	0.24 (0.37) ^†^	0.16 (0.16) ^‡^	196.07 (86.00)
	Internal Oblique	0.42 (0.68)	0.20 (0.22)	185.67 (87.00)
	Transversus Abdominis	2.75 (5.24)	0.79 (0.99)	141.61 (169.00)
	Composite	1.64 *	0.89 *	233.43 *
FF	Anterior Rectus Sheath	8.29 (15.76)	3.45 (4.51)	124.54 (84.00)
	Posterior Rectus Sheath	7.31 (67.69)	3.02 (14.04)	133.33 (190.00)
	Linea Alba	3.92 (3.80)	1.40 (2.88)	225.44 (138.00)
	Peritoneum	4.42 (14.15)	2.24 (3.66)	135.71 (108.00)
	Rectus Abdominis	0.30 (0.88)	0.14 (0.32)	135.60 (152.00)
	External Oblique	0.57 (1.27)	0.28 (0.48)	215.06 (243.00)
	Internal Oblique	0.58 (1.11)	0.40 (0.41)	185.46 (127.00)
	Transversus Abdominis	0.81 (2.44)	0.49 (0.94)	182.43 (118.00)
	Composite	1.12 (2.79)	0.53 (1.22)	195.55 (68.00)

* Sample size is 3. ^†^ U = 196.00, z = 2.62, *p* = 0.009. ^‡^ U = 191.00, z = 2.43, *p* = 0.015. ^§^ U = 46.00, z = 2.46, *p* = 0.014.

## Data Availability

The data presented in this study are available on request from the corresponding author.

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
