# Peer review of "Mechanical Characterization of the Human Abdominal Wall Using Uniaxial Tensile Testing"

_bioengineering, 2023, doi:10.3390/bioengineering10101213_

Round 1
Reviewer 1 Report
It was a study about the evaluation of mechanical properties of the human abdominal wall via utilizing the uniaxial tension test on fresh-frozen and fresh-never-frozen cadavers. Here are some comments on this study that should be considered before publication:
1- Please check the number of Figures and Tables again. There are mistakes in their numbering.
2- The quality of images should be improved.
3- Please add references related to the line number 387.
There are some grammatical mistakes in the text that should be corrected. Some of them are as follows:
- … available in the literature, however, we can ascertain …
- … this study would have likely had similar results to Cardosa [15] had they been reported.
- …
Author Response
Dear reviewer,
The authors thank you for taking the time to review our paper entitled "Mechanical characterization of the human abdominal wall using uniaxial tensile testing." Please see the attachment containing our point-by-point response to your comments.
Sincerely,
The Authors

Reviewer 2 Report
-This manuscript described the tensile testing of Human Abdominal Wall in the form of sub layered tissue and composite tissue obtained for FNF and FF cadaver. The content was generally a data generation paper. Although the authors attempted to link the importance of this study to the use of abdominal laparoscopic procedures, the linkage was unclear. What is the advantage of knowing the properties of each tissue layer compared to the whole or composite tissue that could be found elsewhere? this should be addressed for the merit of the study.
-Introduction lacked a review of previous studies of the mechanical properties of Human Abdominal Wall or other similar tissues and their usefulness in abdominal laparoscopic procedures. Large mount of content was about the structure of the Abdominal Wall and abdominal laparoscopic procedures which could be shortened. This should be revised.
-In addition, I do not think there was an actual analysis of the difference in the additive sum of the tissue layered properties and the property of the composite tissue in this study. If so, the use of the composite rule of mixture for each layer or others might be calculated and compared with composite properties.
-Discussion should also be revised to address more on the applicability of such data in this study. At the moment, the only comparison between data in this manuscript and previously reported data was discussed which is of low interest. It is known that the testing conditions and variations in tissue sources would result in the discrepancy of the data.
-Numbers of figures and tables must be rechecked for correctness.
Other specific comments can be found in the attached file.

Author Response

(The authors gave the same response as above.)

Round 2
Reviewer 2 Report
NA